# Psoriasis in Patients Attending a Large HIV Clinic in Trinidad

**DOI:** 10.3390/medsci10010009

**Published:** 2022-01-29

**Authors:** Robert Jeffrey Edwards, Leon Omari Lavia, Jonathan Edwards, Gregory Boyce

**Affiliations:** 1Medical Research Foundation of Trinidad and Tobago, 7 Queen’s Park East, Port of Spain, Trinidad and Tobago; dataanalyticstt1@gmail.com (L.O.L.); jonathan.r.a.edwards@gmail.com (J.E.); greg.boyce@gmail.com (G.B.); 2Department of Paraclinical Sciences, University of the West Indies, St. Augustine, Trinidad and Tobago

**Keywords:** psoriasis, HIV, clinic, prevalence

## Abstract

The data on psoriasis in persons infected with HIV in the Caribbean is sparse. A cross-sectional study was conducted on patients attending the HIV Clinic diagnosed with psoriasis where sociodemographic data and data on the pattern of psoriasis were collected and analysed using SPSS version 25. Over the period April 2002–December 2018, 37 persons attending the HIV clinic were diagnosed with psoriasis, age range at psoriasis diagnosis 13–70 years, mean age at diagnosis 37.7 years and 32 (86.5%) were male. Over the study period, 8916 patients were registered at the HIV Clinic and the prevalence of psoriasis among these patients was 0.42% which is less than the reported prevalence of psoriasis among persons infected with HIV of 2–3% in other studies. Severe/generalised psoriasis was present in 7 (18.9%) patients while 30 (81.1%) patients had mild/moderate psoriasis. A family history of psoriasis was present in 1 (2.7%) patient, psoriatic arthritis was present in 2 (5.4%) patients and 31 (83.8%) patients reported that there was improvement in the psoriasis with the topical therapy provided. The study makes an original contribution in the determination of the prevalence and pattern of psoriasis among patients attending a HIV Clinic in Trinidad.

## 1. Introduction

Psoriasis is a chronic, systemic, immune-mediated disease with predominantly skin and joint manifestations [1] and evidence suggests that psoriasis may be associated with important systemic consequences [2,3,4]. Severe psoriasis has been linked to a number of co-morbidities including psoriatic arthritis, metabolic syndrome, diabetes mellitus, cardiovascular disease and inflammatory bowel disease which has been associated with worse clinical outcomes and can negatively affect the quality of life [2,3,4,5]. Psoriasis has been reported to affect 2–3% of the general population [6,7] ranging globally from 0.09% in Tanzania to 11.4% in Norway [7] with an equal prevalence in males and females [7,8]. In the USA, the prevalence of psoriasis among adults was reported to be 3.2% [5], 2.5% among Caucasians and 1.9% among African-Americans [5] indicating an ethnic difference in the prevalence of psoriasis. 

In persons living with HIV (PLHIV), dermatologic diseases are very common and may be the initial manifestation of HIV infection and are associated with significant morbidity and mortality [9,10]. Psoriasis has been observed throughout the course of HIV infection ranging in severity from mild to severe and may be the first presentation of HIV infection [11,12] and psoriatic arthritis has been described as more common and severe in these patients [10]. The prevalence of psoriasis among PLHIV has been reported to be 2–3% [10,11,12] but other studies have reported prevalence rates of 4–6% in PLHIV [11,13]. HIV infection is associated with a depletion of CD4+ lymphocytes and HIV associated psoriasis tends to present in patients with low CD4+ cell counts and an overall reduced immune status [14]. This seems paradoxical as psoriasis is associated with activation of T cells which produce a wide range of pro-inflammatory mediators that result in keratinocyte activation and proliferation and therapies that reduce T cells are linked to improvement in psoriasis [14].

Trinidad and Tobago (T&T) comprise a single nation and are the southernmost islands of the Caribbean with a population of approximately 1,394,973 persons (2019 mid-year estimate). The ethnic configuration of the population comprises of persons of East Indian descent (35.4%), persons of African origin (34.2%), mixed races (23.8%), and 8.4% of other ethnic groups (Asian, European, Middle Eastern). Persons of African origin came to the Caribbean via the African slave trade to work on plantations, mainly from West and Central Africa (Igbo, Kongo and Malinke people) [15,16]. The East Indians came to Trinidad as indentured labourers following the abolition of slavery [17], mainly from the Uttar Pradesh and Bihar regions of north India, and a smaller number from Bengal and various areas in south India.

The first cases of AIDS in Trinidad and Tobago were reported among gay/bisexual men in 1983 [18] and from 1985, there was a transition to mainly heterosexual transmission of HIV [19]. Antiretroviral therapy (ART) became available in 2002 funded by the government [20] and it is estimated that there are 11,000 persons living with HIV (PLHIV) in Trinidad and Tobago and 27% of these persons are not on (ART) [21].

There are limited data on the prevalence of psoriasis in the Caribbean. Among new patients attending the Skin Clinic at the General Hospital in Port of Spain, Trinidad, Quamina reported a prevalence of psoriasis of 4% in 1975 [22] and Suite reported a prevalence of 5.1% in 2006 [23]. In 1980, La Grenade and Alabi reported a prevalence of psoriasis of 2.3% in new patients attending a Clinic in Jamaica [24]. In the Caribbean, there are no data on the prevalence of psoriasis in the general population and no data on the prevalence of psoriasis in PLHIV. Thus, the purpose of the study is to determine the prevalence and pattern of psoriasis among PLHIV attending a large HIV Clinic in Trinidad.

## 2. Subjects and Methods

The Medical Research Foundation of Trinidad and Tobago (MRFTT) is the largest HIV Treatment and Care Facility in the English speaking Caribbean and as of 31 December 2018, there were 8916 patients ever enrolled in care. As part of the routine standard of care at the HIV Clinic, data are collected on all patients using the pre- designed pro-forma surveillance form and included demographic data, contact information for next of kin/relatives/friends, ethnicity, sexual orientation, education and employment status.

All patients attending the HIV Clinic who had skin conditions were referred to the Dermatology Clinic at the General Hospital, Port of Spain (a few minutes waking distance from MRFTT) for diagnosis/treatment. In addition, patients diagnosed psoriasis who are found to be HIV infected are referred to the HIV Clinic. Patients were reported as having psoriasis if they had a history of a diagnosis of psoriasis by a dermatologist.

The MRFTT has an electronic medical records system named CELLMA. A list of patients diagnosed with psoriasis who attended the HIV Clinic during the period April 2002–December 2018 was generated using CELLMA.

### 2.1. Inclusion Criteria

Age 18 years and olderPatients living with HIVPatients attending the MRFTT with a diagnosis of psoriasis

### 2.2. Exclusion Criteria

Age under 18 years oldHIV seronegative individualsPatients attending the MRFTT who have not been diagnosed with psoriasis

A cross-sectional study among patients diagnosed with psoriasis attending the MRFTT was conducted during the period January–March 2019, where a questionnaire was administered to collect data including sociodemographic information, sexual orientation, date of diagnosis of psoriasis, diagnosis of psoriasis before or after HIV infection, severity of psoriasis, family history of psoriasis, medication used to treat psoriasis, response of psoriasis to treatment and a history of psoriatic arthritis. If the patients were deceased or lost to follow up, data were collected from the next of kin/relatives/friends of patients diagnosed with psoriasis. In addition, data were abstracted from client charts to obtain the laboratory records of patients diagnosed with psoriasis including the CD4 count and HIV viral load at diagnosis of HIV infection.

Severe psoriasis was defined in the study as a body surface area (BSA) involved >10%. [25]. If one considers the outstretched hand, including the thumb and fingers (a ‘handprint’) as corresponding to approximately 1% BSA [26]. Severe psoriasis was defined for the patients as area of involvement totaling more than 10 handprints especially if the face, palms of the hands, soles of the feet or genitals were involved.

### 2.3. Ethical Approval

The study and related study procedures were reviewed and approved by the Campus Research Ethics Committee, University of the West Indies, St Augustine, Trinidad, approval number CEC534/03/18.

### 2.4. Data Collection and Data Analysis

All data collected on the structured surveillance forms and from the medical records were analysed using IBM Statistical Package for Social Science (SPSS) Version 25.

The prevalence of psoriasis was determined by dividing the number of patients who had been given a diagnosis of psoriasis by a dermatologist by the number of people in the clinic population during the study time period. Any missing data were dropped from the statistical analysis. Numerical variables were summarized using the mean, median, standard deviation and range and the means compared using the Independent Samples T test. Chi square tests (X^2^) or Fisher’s Exact Test were used as appropriate to examine differences in categorical variables by ethnicity, year of diagnosis of psoriasis, sexual orientation (heterosexual vs. homosexual/bisexual), diagnosis of psoriasis before or after diagnosis of HIV infection. severity of psoriasis, family history of psoriasis, medication used to treat psoriasis, response of psoriasis to treatment, history of psoriatic arthritis and the diagnosis of psoriasis. Bivariate logistic regression analyses were conducted to examine associations between the independent variables and the diagnosis of psoriasis. The results were presented as odds ratios (95% CI).

## 3. Results

As of 31 December 2018, 8916 patients were registered at the HIV Clinic, MRFTT. There were 6565 (73.6%) patients who were currently engaged in active follow up, 1603 (18.0%) deaths, and 748 (8.4%) were lost to follow up. The ethnic composition of persons attending the HIV clinic consisted of 5322 (59.7%) persons of African origin, 692 (7.8%) persons of East Indian origin, 2721 (30.5%) persons of mixed race and 67 (0.75%) other persons (Caucasian, Chinese, Syrian/Lebanese). No data were available on ethnicity for 114 (1.3%) persons.

Over the period April 2002–December 2018, 37 persons attending the HIV Clinic were diagnosed with psoriasis, age range at diagnosis 13–70 years, mean age at diagnosis 37.7 years, 32 (86.5%) were male with a male: female ratio of 6.4:1. Severe/generalised psoriasis was present in 7 (18.9%) of patients while 30 (81.1%) of patients had mild/moderate psoriasis. A family history of psoriasis was present in 1 (2.7%) patient, psoriatic arthritis was present in 2 (5.4%) patients and 31 (83.8%) reported that there was improvement in the psoriasis with the therapy provided (Table 1).

The prevalence of psoriasis in patients with HIV infection at the HIV Clinic, MRFTT was 0.42%. The prevalence of psoriasis among those of African origin was 0.40%, those of East Indian origin 0.58% and those of mixed race 0.44%.

There were no statistically significant associations between the independent variables and the diagnosis of psoriasis using bivariate logistic regression analyses as the number of patients with a diagnosis of psoriasis was small.

Data were available on 27 (73%) patients about the time between the diagnosis of psoriasis and HIV. A diagnosis of psoriasis before a diagnosis of HIV infection was reported in 17 (63%) of 27 patients and psoriasis was the presenting manifestation of HIV infection in 2 (7.4%) patients, one with severe psoriasis and one with mild/moderate psoriasis. A diagnosis of psoriasis after a diagnosis of HIV infection was reported in 10 (37%) of the study patients. There were no statistically significant associations between the independent variables and the diagnosis of psoriasis before or after HIV infection.

“Treat all” where all patients are treated with antiretroviral therapy (ART) regardless of CD4 cell counts was started in 2017 at the HIV Clinic and all 31 patients diagnosed with psoriasis attending the MRFTT who were alive during the study were on ART.

## 4. Discussion

The Medical Research Foundation of Trinidad and Tobago (MRFTT) had 6565 patients living with HIV (PLHIV) in active care and follow up as of 31 December 2018 and this represents approximately 70% of the PLHIV currently enrolled in care in Trinidad and Tobago. Of the 37 patients diagnosed with psoriasis in our study, there was a male predominance (M:F = 6.4:1), this was supported in the study by Suite [23] who conducted a retrospective analysis of the prevalence of psoriasis among new patients attending a dermatology clinic in Trinidad and found psoriasis was more common in males [23]. However, most of the global data showed that psoriasis occurs equally in males and females [7], though some studies showed a slight male preponderance, but these were not statistically significant [7]. Psoriasis can occur at any age, in our study, the mean age at onset of psoriasis was 37.7 years which was similar to a mean age of onset of 33 years reported in some studies [7,27], however Suite [23] showed that the peak presentation occurred among the age group of 50–59 years. Other studies suggested that there were two peaks of the disease, the first occurring between 16–22 years and the second between 57–60 years of age indicating a bimodal onset [7,28], however this was not seen our study where the number of cases of psoriasis were small.

The prevalence of psoriasis in patients with HIV infection attending the MRFTT was 0.42% which is less than the reported prevalence of psoriasis among PLHIV of 2–3% [10,11,12]. A study of 2000 HIV infected men in San Francisco showed a prevalence of psoriasis of 2.5% [29], these patients were seen in a tertiary care centre, most were Caucasians and almost all were MSM, so the data may not be applicable to other HIV infected populations. Another study in Berlin of 700 HIV infected patients reported a prevalence of 5%, almost all the patients were Caucasians [30]. Studies in the USA have shown there are ethnic differences in the prevalence of psoriasis where the prevalence was higher among Caucasians than among African-Americans [5,31]. In our study, the prevalence of psoriasis among those patients of African origin was 0.40% which is comparable to the prevalence of psoriasis in the West African populations of Nigeria (0.5%) [32], Sierra Leone (<0.2%) [33] and Senegal (0.6%) [34] which suggests a similar genetic ancestry as persons of African origin came to T&T via the African slave trade mainly from West Africa [16]. However, the prevalence of psoriasis among patients of East Indian origin was 0.58% indicating there were no statistically significant ethnic differences (*p* = 0.74) in the prevalence of psoriasis in our study. In India, the prevalence of psoriasis varies from 0.44% to 2.8% [35].

Studies have reported the clinical course of psoriasis in patients with HIV infection to be more severe and progressive with atypical manifestations and frequently non-responsive to standard treatment [10,11,12]. In our study, severe/generalised psoriasis defined as psoriasis affecting greater than 10% of the body surface area was present in 7 (18.9%) patients while 30 (81.1%) patients had mild/moderate psoriasis, which was similar to the findings by Yeung et al. [36] in a population-based study which reported than in 8747 patients with psoriasis in the UK, 51.8% had mild psoriasis, 35.8% moderate psoriasis and 12.4% has severe psoriasis. Psoriatic arthritis was present in 2 (5.4%) patients in our study but has been reported to affect 23–50% of HIV infected patients with psoriasis [11,37]. HIV associated psoriasis has been reported to have high failure rates to the regularly prescribed medications [38], nonetheless, in our study, 6 (16.2%) patients reported that there was no improvement in the psoriasis with the therapy provided including two patients with severe/generalised psoriasis who were treated with methotrexate. However, 31 (83.8%) patients reported there was improvement in their psoriasis with the topical therapy provided. “Treat all” where all patients are treated with antiretroviral therapy (ART) regardless of CD4 cell counts was started in 2017 and all patients diagnosed with psoriasis who are in active care and follow up at the MRFTT are on ART which results in reconstitution of the immune system and may account for the improvement in response to therapy for psoriasis in HIV infected patients attending our clinic.

The treatment of psoriasis in patients with HIV infection is therapeutically challenging as this represents a T cell mediated disease in an environment of a depletion of T cells [14]. The majority of the systemic agents used in the treatment of psoriasis are immunosuppressive and can potentially result in severe complications in patients with HIV infection and adverse drug interactions in patients on ART [10,14]. There are no randomized clinical trials to guide treatment in these patients and the recommended initial therapy for mild to moderate psoriasis should include ART, topical therapy and phototherapy with oral retinoids as second line treatment [10]. For refractory psoriasis, careful treatment with systemic immunosuppressive therapy including biologics should be considered and patients should be closely monitored for potential adverse events [10,38].

There are some limitations to the study, the small number of patients diagnosed with psoriasis affected data analysis. Before ART became widely available, some patients may have died with AIDS before a diagnosis of psoriasis was made resulting in an underestimate in psoriasis diagnoses in the clinic population. Six patients were deceased and data on the pattern of psoriasis and response to therapy had to be obtained from relatives with its inherent bias which may affect the accuracy of data.

## 5. Conclusions and Future Recommendations

This is the first study from the Caribbean to determine the prevalence and pattern of psoriasis in HIV infected patients, however due to the relatively low prevalence of psoriasis at the HIV clinic (and the small number of patients with psoriasis in the study), it may be difficult to generalize our study results to psoriasis among HIV infected patients in other settings. Further studies on the epidemiology of psoriasis among PLHIV in the Caribbean are required.

## Figures and Tables

**Table 1 medsci-10-00009-t001:** Baseline characteristics of the Study Population (*n* = 37).

Descriptive Statistic	Total (*n* = 37)
Mean Age at diagnosis of psoriasis	37.7 years
Age Range at diagnosis of psoriasis	13–70 years
Age group (years) at diagnosis of psoriasis	
<19	3 (8.1%)
20–29	6 (16.2%)
30–39	14 (37.8%)
40–49	10 (27.0%)
50–59	2 (5.4%)
60+	2 (5.4%)
**Sex**	
Males	32 (86.5%)
Females	5 (13.5)
Ethnicity	
African	21 (56.8%)
East Indian	4 (10.8%)
Mixed	12 (32.4%)
Current status	
Alive	31 (83.8%)
Deceased	6 (16.2%)
Sexual Orientation	
Heterosexual	(78.4%)
Homosexual/Bisexual	8 (21.6%)
Severity of psoriasis	
Mild/moderate	30 (81.1%)
Severe/generalised	7 (18.9%)
Psoriatic arthritis	
Yes	2 (5.4%)
No	35 (94.6%)
Family history of psoriasis	
Yes	1 (2.7%)
No	36 (97.3%)
Medication used to treat psoriasis	
Topical therapy	35 (94.6%)
Systemic therapy and Topical therapy	2 (5.4%)
Reported improvement with the response to the therapy for psoriasis	
Yes	31 (83.8%)
No	6 (16.2%)
Median CD4 count at HIV diagnosis (cells/mm^3^)	167
Median HIV viral load at HIV diagnosis (copies/mL)	122,000

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
