# Peer review of "Psoriasis in Patients Attending a Large HIV Clinic in Trinidad"

_medsci, 2022, doi:10.3390/medsci10010009_

Round 1

Reviewer 1 Report

Dear Authors

Thank you, Robert Jeffrey Edwards, Leon Omari Lavia, Jonathan Edwards, Gregory Boyce and best wishes for your valuable work entitled “Psoriasis in patients attending a large HIV Clinic in Trinidad”. I appreciate for your efforts and research manuscript. However, I would like to share my comments to improve your manuscript as well.

Comments to authors

Section

Line

Page

Comments

Abstract

-

1

Please mention the sample size as it is not clearly described in abstract.

Data collection and data analysis

99

3 of 7

Please include “structured surveillance forms” in this section of deposit as supplementary materials.

…….

114

3 of 7

Please check and separate the Results section.

Results

134

4 of 7

-----Please add the “bivariate logistic regression analyses” r2 values in table 1.

………

..

5 of 7

Please add the new section as “conclusion and future recommendation”

Author Response

Reviewer #1

Please mention the sample size as it is not clearly described in abstract.

Answer

Over the period April 2002 – December 2018, 37 persons attending the HIV clinic were diagnosed with psoriasis, age range at psoriasis diagnosis 13-70 years, mean age at diagnosis 37.7 years and 32 (86.5%) were male. Over the study period, 8,916 patients were registered at the HIV Clinic and the prevalence of psoriasis among these patients was 0.42%.

Please include “structured surveillance forms” in this section of deposit as supplementary materials.

Answer

The structured surveillance form and the questionnaire were included as supplementary materials

Please check and separate the Results section.

Answer

A separate results section was added

Please add the “bivariate logistic regression analyses” r2 values in table 1.

Answer

Table 1: Baseline characteristics of the Study Population (n=37)

Descriptive Statistic

Total (n=37)

r2                   p value

Mean Age at diagnosis of psoriasis

37.7 years

-0.086               0.630

Age Range at diagnosis of psoriasis

13-70 years

Age group (years) at diagnosis of psoriasis

<19

20-29

30-39

40-49

50-59

60+

3 (8.1%)

6 (16.2%)

 14 (37.8%)

10 (27.0%)

2 (5.4%)

2 (5.4%)

-0.208               0.121

Sex

Males

Females

32 (86.5%)

5 (13.5)

-.0213               0.196

Ethnicity

African

East Indian

Mixed

21 (56.8%)

4 (10.8%)

12 (32.4%)

0.098                0.522

Current status

Alive

Deceased

31 (83.8%)

6 (16.2%)

0.197                0.236

Sexual Orientation

Heterosexual

Homosexual/Bisexual

29   (78.4%)

8 (21.6%)

0.254                0.291

Severity of psoriasis

Mild/moderate

Severe/generalised

30 (81.1%)

7 (18.9%)

N/A

Psoriatic arthritis

Yes

No

2 (5.4%)

35 (94.6%)

-0.614               0.055

Family history of psoriasis

Yes

No

1 (2.7%)

36 (97.3%)

0.083                0.618

Medication used to treat psoriasis

Topical therapy

Systemic therapy and Topical therapy

35 (94.6%)

2 (5.4%)

0.190                0.101

Reported improvement with the response to the therapy for psoriasis

Yes

No

31 (83.8%)

6 (16.2%)

0.213                 0.196

Median CD4 count at HIV diagnosis (cells/mm3 )

167

-0.041               0.816

Median HIV viral load at HIV diagnosis (copies/ml)

122,000

0.087                0.654

Put in the r2 and p values in the Table but all are statistically insignificant. Since most Clinicians tend to be confused by r2, it was decided to leave them out in the manuscript submitted for publication

Please add the new section as “conclusion and future recommendation”

Answer

New section “Conclusion and future recommendation” added

Reviewer 2 Report

The presented retrospective analysis is clinically important because for the first time from the Caribbean. The authors determined the prevalence and pattern of psoriasis in HIV infected patients. Interestingly the prevalence was much lower than in the general population.  IT has been however stated that that due to small number of the patients the results cannot be conclusive and further studies on the epidemiology of psoriasis among PLHIV in the Caribbean are required.

The conducted analysis of the psoriasis patients attending the HIV Clinic in  the Caribbean is sparse. Although the weak side of the study is the sample size was quite small - 37 persons with psoriasis, it still highlights the clinical relevance of the study which on the other hand points to the strong side.  Interestingly, the authors noted much smaller prevalence of psoriasis among their HIV patients (0.42%) comparing to the healthy population and other similar studies as well (2-3%). The authors could more in details discuss this difference in the discussion section. Further larger cohort studies are needed and the authors should be encouraged to conduct in order to precisely  determine the prevalence and pattern of psoriasis among patients attending a HIV Clinic in Trinidad.

Author Response

Reviewer #2

Interestingly, the authors noted much smaller prevalence of psoriasis among their HIV patients (0.42%) comparing to the healthy population and other similar studies as well (2-3%). The authors could more in details discuss this difference in the discussion section.

Answer

The prevalence of psoriasis in patients with HIV infection attending the MRFTT was 0.42% which is less than the reported prevalence of psoriasis among PLHIV of 2-3% [10, 11, 12]. A study of 2000 HIV infected men in San Francisco showed a prevalence of psoriasis of 2.5% (Obush), these patients were seen in a tertiary care centre, most were Caucasians and many were MSM, so the data may not be applicable to other HIV infected populations.  Another study in Berlin of 700 HIV infected patients reported a prevalence of 5%, most patients were Caucasians.  Studies in the USA have shown there are ethnic differences in the prevalence of psoriasis where the prevalence was higher among Caucasians than among African-Americans [5, 25]. In our study, the prevalence of psoriasis among those patients of African origin was 0.40% which is comparable to the prevalence of psoriasis in the West African populations of Nigeria (0.5%) [26], Sierra Leone (<0.2%) [27] and Senegal (0.6%) [28] which suggests a similar genetic ancestry as persons of African origin came to T&T via the African slave trade mainly from West Africa [14]. However, the prevalence of psoriasis among patients of East Indian origin was 0.58% indicating there were no statistically significant ethnic differences (p=0.74) in the prevalence of psoriasis in our study. In India, the prevalence of psoriasis varies from 0.44% to 2.8% [29].

Reviewer 3 Report

I read the article entitled “Psoriasis in patients attending a large HIV Clinic in Trinidad”.

The data is interesting however some concerns should be clarified and revised.

Abstract section: Well-written

Introduction section: Authors should give detailed information about the prevalence of psoriasis in the Caribbean. The following sentence is not clear: “There is very little data on the prevalence of psoriasis in the Caribbean”. They may change this sentence as “according to restricted data, the prevalence of psoriasis in the Caribbean is …….”

Furthermore, authors should discuss the prevalence of psoriasis in HIV patients and the underlying etiopathology in these patients (corresponding to a low CD4+ count and an overall decreased immune status) (paragraph 2, line 36-41).

Methods: Authors declared that if the patients were deceased or lost to follow-up, data were collected from the next of kin/relatives/friends of patients diagnosed with psoriasis. I think this is an important limitation. Information learned from the patient’s relatives may include bias.

How do they classify patients as severe?

Typos should be corrected.

Results: The data about the time between the diagnosis of psoriasis and HIV is not clear. Furthermore, psoriasis may be the first disease manifestation of HIV. Are there any patients presenting with psoriasis and then diagnosed with HIV?

Please add the data about antiretroviral therapy.

Discussion: Authors reported that the prevalence of psoriasis in patients with HIV infection attending the MRFTT was 0.42% which is less than the reported prevalence of psoriasis among PLHIV of 2-3%. The authors should discuss the probable factors for this difference.

Add a paragraph about the difficulties of immunosuppressive therapy in HIV patients.

Author Response

Reviewer #3

Introduction section: Authors should give detailed information about the prevalence of psoriasis in the Caribbean. The following sentence is not clear: “There is very little data on the prevalence of psoriasis in the Caribbean”. They may change this sentence as “according to restricted data, the prevalence of psoriasis in the Caribbean is …….”

Answer

There are limited data on the prevalence of psoriasis in the Caribbean, among new patients attending the Skin Clinic at the General Hospital in Port of Spain, Trinidad, Quamina reported a prevalence of psoriasis of 4% in 1975 [ ] and Suite reported a prevalence of 5.1% in 2006 [ ]. In 1980, La Grenade and Alabi reported a prevalence of psoriasis of 2.3% in new patients attending a Clinic in Jamaica. In the Caribbean, there are no data on the prevalence of psoriasis in the general population and no data on the prevalence of psoriasis in PLHIV.

Furthermore, authors should discuss the prevalence of psoriasis in HIV patients and the underlying etiopathology in these patients (corresponding to a low CD4+ count and an overall decreased immune status) (paragraph 2, line 36-41).

Answer

The prevalence of psoriasis among PLHIV has been reported to be 2-3% but other studies have reported prevalence rates of 4%-6% in PLHIV [11, 22]. HIV infection is associated with a depletion of CD4+ lymphocytes and HIV associated psoriasis tends to present in patients with a low CD4+ cell counts and an overall reduced immune status. This seems paradoxical as psoriasis is associated with activation of T cells which produce a wide range of pro-inflammatory mediators that result in keratinocyte activation and proliferation and therapies that reduce T cells are linked to improvement in psoriasis.

Methods: Authors declared that if the patients were deceased or lost to follow-up, data were collected from the next of kin/relatives/friends of patients diagnosed with psoriasis. I think this is an important limitation. Information learned from the patient’s relatives may include bias.

This is included in the limitations of the study

How do they classify patients as severe?

Answer

Severe psoriasis was defined in the study as a body surface area (BSA) involved > 10%.  If one considers the outstretched hand, including the thumb and fingers (a 'handprint') as corresponding to approximately 1% BSA. Severe psoriasis was defined for the patients as area of involvement totalling more than 10 handprints especially if the face, of t hands, soles of the feet or genitals were involved.

Typos should be corrected

Answer

The typos were corrected

Results:

The data about the time between the diagnosis of psoriasis and HIV is not clear. Furthermore, psoriasis may be the first disease manifestation of HIV. Are there any patients presenting with psoriasis and then diagnosed with HIV?

Answer

Data were available on 27 (73%) patients about the time between the diagnosis of psoriasis and HIV. A diagnosis of psoriasis before a diagnosis of HIV infection was reported in in 17 (63%) of 27 patients and psoriasis was the presenting manifestation of HIV infection in 2 (7.4%) patients, one with severe psoriasis and one with mild/moderate psoriasis. A diagnosis of psoriasis after a diagnosis of HIV infection was reported in 10 (37%) of the study patients. There were no statistically significant associations between the independent variables and the diagnosis of psoriasis before or after HIV infection using bivariate logistic regression analyses.

Please add the data about antiretroviral therapy.

Answer

“Treat all” where all patients are treated with antiretroviral therapy (ART) regardless of CD4 cell counts was started in 2017 at the HIV Clinic and all 31 patients diagnosed with psoriasis attending the MRFTT who were alive during the study were on ART

Discussion:

Authors reported that the prevalence of psoriasis in patients with HIV infection attending the MRFTT was 0.42% which is less than the reported prevalence of psoriasis among PLHIV of 2-3%. The authors should discuss the probable factors for this difference.

Answer

The prevalence of psoriasis in patients with HIV infection attending the MRFTT was 0.42% which is less than the reported prevalence of psoriasis among PLHIV of 2-3% [10, 11, 12]. A study of 2000 HIV infected men in San Francisco showed a prevalence of psoriasis of 2.5%, these patients were seen in a tertiary care centre, most were Caucasians and many were MSM, so the data may not be applicable to other HIV infected populations.  Another study in Berlin of 700 HIV infected patients reported a prevalence of 5%, most patients were Caucasians.  Studies in the USA have shown there are ethnic differences in the prevalence of psoriasis where the prevalence was higher among Caucasians than among African-Americans [5, 25]. In our study, the prevalence of psoriasis among those patients of African origin was 0.40% which is comparable to the prevalence of psoriasis in the West African populations of Nigeria (0.5%) [26], Sierra Leone (<0.2%) and Senegal (0.6%) which suggests a similar genetic ancestry as persons of African origin came to T&T via the African slave trade mainly from West Africa [14]. However, the prevalence of psoriasis among patients of East Indian origin was 0.58% indicating there were no statistically significant ethnic differences (p=0.74) in the prevalence of psoriasis in our study. In India, the prevalence of psoriasis varies from 0.44% to 2.8%.

Add a paragraph about the difficulties of immunosuppressive therapy in HIV patients.

Answer

The treatment of psoriasis in patients with HIV infection is therapeutically challenging as this represents a T cell mediated disease in an environment of a depletion of T cells. The majority of the systemic agents used in the treatment of psoriasis are immunosuppressive and can potentially result in severe complications in patients with HIV infection and adverse drug interactions in patients on ART. There are no randomized clinical trials to guide treatment in these patients and recommended initial therapy for mild to moderate psoriasis should include ART, topical therapy, phototherapy with oral retinoids as second line treatment.  For refractory psoriasis, careful treatment with systemic immunosuppressive therapy including biologics should be considered and patients should be closely monitored for potential adverse events.

Round 2

Reviewer 3 Report

I thank the authors. They replied and corrected all concerns, properly. The revised version of the manuscript is ready for publication.